# Extracorporeal Shock Wave Enhances the Cisplatin Efficacy by Improving Tissue Infiltration and Cellular Uptake in an Upper Urinary Tract Cancer Animal and Human-Derived Organoid Model

**DOI:** 10.3390/cancers13184558

**Published:** 2021-09-11

**Authors:** Hao-Lun Luo, Hui-Ying Liu, Yin-Lun Chang, Yu-Li Su, Chun-Chieh Huang, Xin-Jie Lin, Yao-Chi Chuang

**Affiliations:** 1Department of Urology, Kaohsiung Chang Gung Memorial Hospital and Chang Gung University College of Medicine, Kaohsiung 83301, Taiwan; alesy@cgmh.org.tw (H.-L.L.); ying1011@cgmh.org.tw (H.-Y.L.); tailanylyl@cgmh.org.tw (Y.-L.C.); cololoninacy@cgmh.org.tw (X.-J.L.); 2Center for Shockwave Medicine and Tissue Engineering, Kaohsiung Chang Gung Memorial Hospital and Chang Gung University College of Medicine, Kaohsiung 83301, Taiwan; 3Department of Hematology and Oncology, Kaohsiung Chang Gung Memorial Hospital and Chang Gung University College of Medicine, Kaohsiung 83301, Taiwan; yolisu@cgmh.org.tw; 4Department of Radiation Oncology, Kaohsiung Chang Gung Memorial Hospital and Chang Gung University College of Medicine, Kaohsiung 83301, Taiwan; b8705050@cgmh.org.tw

**Keywords:** shock wave, chemotherapy, upper urinary tract urothelial carcinoma, patient-derived organoid

## Abstract

**Simple Summary:**

Chemotherapy is the standard treatment for advanced upper urinary tract urothelial carcinoma (UTUC), and shock wave treatment is a common strategy for upper urinary tract stones. We explored the mechanism of the combination therapy of low-energy shock waves (LESWs) and cisplatin for UTUC in vitro, in vivo, and in patient-derived organoid (PDO) models. Histopathological examination showed more deteriorated cell arrangement and oedema in the combination treatment group than in the cisplatin only group. Immunohistochemical analysis revealed reduced expression of proliferation markers, increased expression of apoptosis markers, and increased cisplatin infiltration in the combination treatment group. Western blotting revealed decreased cisplatin efflux and membranous protein levels after shock wave application. Moreover, LESW improved the cytotoxicity of cisplatin in the preclinical PDO model of UTUC. Our findings showed that LESW enhanced the antitumour efficacy of cisplatin in UTUC. Hence, combination therapy could have promising applications for locally advanced UTUC in clinical settings.

**Abstract:**

Upper urinary tract urothelial carcinoma (UTUC) is a relatively rare cancer with a poor prognosis if diagnosed at an advanced stage. Although cisplatin-based chemotherapy is a common treatment strategy, it has a limited response rate. Shock wave lithotripsy is a common treatment for upper urinary tract stones. Low-energy shock waves (LESWs) temporarily increase tissue permeability and enhance drug penetration to the targeted tissue. However, no study has investigated the efficacy of the combination of shock wave lithotripsy and chemotherapy in UTUC. Hence, in this study, we aimed to identify the potential application of the combination of LESW and chemotherapy in UTUC. We evaluated the synergistic effects of LESW and cisplatin in vitro, in vivo, and in patient-derived organoid (PDO) models. Compared with cisplatin alone, the combination treatment caused more significant tumour suppression in vitro and in animal models, without increased toxicity. Histological examination showed that compared with animals treated with cisplatin alone, those who received the combination treatment showed more deteriorated cell arrangement and cell oedema. Moreover, LESW improved the cytotoxicity of cisplatin in the preclinical PDO model of UTUC. Thus, LESW combined with cisplatin is a potential new antitumour strategy for improving the treatment response in locally advanced UTUC.

## 1. Introduction

Upper urinary tract urothelial carcinoma (UTUC) is a relatively rare cancer with a poor prognosis if diagnosed at an advanced stage [1]. Renal insufficiency is commonly associated with UTUC, and renal function often worsens after radical nephroureterectomy [2,3]. In clinical practice, cisplatin-based chemotherapy is currently the most effective agent for advanced urothelial cancer. However, renal insufficiency renders the patient ineligible for chemotherapy, and physicians often shift the treatment strategy from cisplatin-based regimens to carboplatin- or paclitaxel-based regimens, which have lower efficacy [4,5]. In addition, chemotherapy is often used as a neoadjuvant treatment strategy. However, it has a limited response rate [6]. Therefore, improving the efficacy of and response to chemotherapy is a clinical challenge.

Shock wave lithotripsy is one of the common treatments for upper urinary tract stones [7]. The technique of target positioning on the upper urinary tract is mature in this treatment. The beneficial effects of low-energy shock waves (LESWs) on cancer have been reported; LESW aids in cancer treatment by increasing the uptake of the chemotherapeutic agent, mainly in vitro [8]. The efficacy of shock wave lithotripsy can even be optimised specifically for cancer cells while sparing normal cells [9]. We hypothesised that the combination of LESW and systemic chemotherapy is reasonable for clinical translation, especially for UTUC located in the most commonly approached anatomic region of shock waves.

The lack of clinical tumour models hinders the innovation of effective therapies for UTUC. Recently, several studies [10,11,12] have reported the value of three-dimensional organoid culture systems for fundamental and translational cancer research. Patient-derived organoids (PDOs) realistically recapitulate the key elements of the disease, including tumour morphology [13], sequential mutation [14], and heterogenous cellular populations analogous to the heterogeneous composition of the native tumour tissue [15]. Additionally, PDO models can accurately reflect the treatment response observed in patients [11,16,17]. Therefore, through this study, we aimed to identify the potential role of LESW in increasing the efficacy of cisplatin in vitro, in vivo, and in preclinical PDO models.

## 2. Materials and Methods

### 2.1. Cell Culture and Reagents

The BFTC909 cell line (a cell line from a patient with cell carcinoma of the renal pelvis) was purchased from Bioresource Collection and Research Center (BCRC). The BFTC909 cell line has been previously described in detail [18]. The UM-UC-14 cell line was purchased from the European Collection of Authenticated Cell Cultures (ECACC). UM-UC-14 cells were cultured in Eagle’s Minimum Essential Medium containing 2 mM glutamine (Thermo Fisher Scientific, Waltham, MA, USA), 1% nonessential amino acids (Thermo Fisher Scientific, Waltham, MA, USA), and 10% heat-inactivated FBS (Thermo Fisher Scientific, Waltham, MA, USA). Cisplatin (1 mg/mL; Kemoplat^®^) was purchased from Fresenius Kabi Oncology Limited, Solan, India. The fluorophores Lucifer Yellow CH dilithium salt (cat# L0259) and Calcein (cat# C0875) were acquired from Sigma, St. Louis, MO, USA. ON-TARGETplus control Non-targeting pool siRNAs (Cat# D-001810-05), ON-TARGETplus Human ZO-1 siRNA (Cat# L-007746-00-0005) and ON-TARGETplus Human E-cadherin siRNA (Cat# L-003877-00-0005) were purchased from Dharmacon, Lafayette, CO, USA. These siRNAs were dissolved at 10 μM in DNase and RNase- free water and stored in 10 μL aliquots at −80 °C until use.

### 2.2. Extracorporeal Shock Wave Exposure In Vitro

BFTC909 cells were transferred to 15-mL polypropylene tubes and spun down at 300 g for 5 min. The bottom of the tubes was covered with ultrasound transmission gel. The EvoTron™ shock wave applicator was gently placed directly on the bottom of the tube. The shock wave frequency was 4 pulses per second. Cells (5 × 10^5^ cells/mL in 0.5 mL) were exposed to 0, 50, 100, 150, 200, 250, 300, 350, 400, 450, 500, 800, or 1600 shock wave pulses at 0.05, 0.12, or 0.25 mJ/mm^2^. The number of viable cells was determined through the trypan blue dye exclusion assay using a haemocytometer. Data were analysed using GraphPad Prism 8.0.2 software, and the results were determined by applying nonlinear regression (curve fit) and the equation inhibitor versus normalised response (variable slope).

### 2.3. Xenograft Tumour Model and Treatment Protocol

All animal procedures were performed according to an IACUC-approved protocol (IACUC number: 2018033001 and 2019030401). Six-week-old SCID (BioLASCO, Taiwan) mice were maintained in microisolators under specific pathogen-free conditions. For transplantation in vivo, 5 × 10^6^ BFTC909 cells or 2.5 × 10^6^ UM-UC-14 cells were injected subcutaneously into the backs of SCID mice for tumour development. The size of the transplanted tumours was measured using vernier callipers twice a week, and the tumour volume was calculated using the following formula: V = 1/2 × (length × width^2^). The values were calculated as the average of the individual relative tumour volumes (relative tumour volume = V_x_/V_0_; V_x_ = volume on Day x and V_0_ = volume on Day 0).

When the tumour volumes reached approximately 100–150 mm^3^, the mice were randomised into four groups (control, LESW, cisplatin 1 mg/kg, and cisplatin 1 mg/kg plus LESW). Three mice in each group were intraperitoneally administered cisplatin (1 mg/kg, thrice a week) or PBS. LESW of 100 pulses/point at 0.12 mJ/mm^2^ at four points was applied to the skin surface above the tumour at the beginning of the first, second, third, and fourth weeks. At the end of treatment, the mice were sacrificed, and the tumours were excised, weighed, and photographed.

### 2.4. Immunohistochemistry

Paraffin sections were deparaffinised with xylene and rehydrated with serial grades of alcohol. Epitope retrieval of Ki-67, γ-H2AX, ZO-1, E-cadherin, and MDR1 was conducted in a pH6 epitope retrieval solution (Leica Microsystems, Wetzlar, Germany) in a water bath at 90 °C on a hot plate for 30 min. All subsequent steps were performed using the Novolink Polymer Detection System (Code: RE7280-K; Leica Biosystems, IL, USA), according to the manufacturer’s instructions [18], as follows: (1) peroxide block placement on the slides for 10 min at room temperature; (2) protein block buffer addition to the slides for 30 min at room temperature; (3) overnight incubation at 4 °C with rabbit monoclonal anti-Ki-67 antibody at a dilution of 1:100 (cat# ab16667; Abcam, Cambridge, UK), rabbit polyclonal anti-γ-H2AX antibody at a dilution of 1:2000 (cat# ab11174; Abcam, Cambridge, UK), rabbit polyclonal anti-ZO-1 antibody at a dilution of 1:400 (cat# 61-7300; Invitrogen, Life Technologies, Carlsbad, CA, USA), rabbit monoclonal anti-E-cadherin antibody at a dilution of 1:100 (cat# 3195; Cell Signaling Technology, Danvers, MA, USA), or rabbit monoclonal anti-MDR1 antibody at a dilution of 1:400 (cat# ab170904; Abcam, Cambridge, UK); (4) incubation with the post primary reagent for 30 min at room temperature; (5) Novolink polymer placement on the slides for 30 min at room temperature; (6) colour development with 3,30-diaminobenzidine tetrahydrochloride (DAB) as a chromogen for 5 min at room temperature; and (7) haematoxylin counterstaining for 10 min. The slides were mounted and examined through light microscopy. The percentage of Ki-67 and γ-H2AX signal staining was semi-quantified using the IHC Toolbox plugin in ImageJ software, which could be effectively used to analyse the accuracy of colour detection in DAB-stained samples through immunohistochemistry [19].

Paraffin sections were deparaffinised with xylene and rehydrated with serial grades of alcohol. After inactivation of endogenous peroxidase with 3% hydrogen peroxide for 10 min at room temperature, DNA denaturation was performed with 0.05 M NaOH for 10 min at room temperature. The slides were rinsed twice with PBS and incubated at room temperature for 30 min with blocking buffer consisting of PBS containing 5% bovine serum albumin. Tissue samples were incubated overnight at 4 °C with rat monoclonal anti-cisplatin DNA adduct antibody at a dilution of 1:100 (cat# MABE416; Millipore, Burlington, MA, USA). The samples were rinsed four times with PBS and incubated for 60 min at room temperature with a rat probe (cat# TA00C2; BioTnA Biotech, Kaohsiung, Taiwan) for 2 h at 37 °C. The Novolink polymer was placed on the slides for 30 min at room temperature, and staining was performed using DAB. The slides were then counterstained with haematoxylin for 10 min, and staining was visualised through light microscopy.

### 2.5. Western Blotting Assays

BFTC909 and UM-UC-14 xenograft tumour tissues and cells were lysed in the RIPA buffer with a protease inhibitor mixture (Roche, Basel, Switzerland). For each lane of 10% SDS–PAGE gel, 30 μg protein of tumour tissue or cell lysates were loaded, separated, and subsequently transferred onto Immobilon-P Transfer Membrane (Millipore, Burlington, MA, USA). The membranes were probed with specific antibodies. The primary antibodies were against ZO-1 (cat# 61-7300; Invitrogen, Life Technologies, Carlsbad, CA, USA), E-cadherin (cat# 3195; Cell Signaling Technology, Danvers, MA, USA), caspase-3 (cat# ZRB1221; Sigma, St. Louis, MO, USA), caspase-9 (cat##9502; Cell Signaling Technology, Danvers, MA, USA), and beta-actin (cat# ZRB1312; Sigma, St. Louis, MO, USA). The primary antibodies were used: ZO-1 (1:1000), E-cadherin (1:1000), caspase-3 (1:1000), caspase-3 (1:1000), and beta-actin (1:5000). The secondary antibodies were added and incubated for 2 h and then visualized using chemiluminescence. Enhanced chemiluminescence (ECL) western blotting reagents were obtained from Pierce Biotechnology (Rockford, IL, USA).

### 2.6. RNA Isolation and Real-Time PCR

RNA was extracted by means of QIAGEN RNA purification kit from BFTC909 and UM-UC-14 xenograft tumour tissues. One microgram RNA of each sample will be reverse transcribed using RevertAidTM H Minus Reverse Transcriptase (Fermentas, Waltham, MA, USA). Real-time PCR was performed using SYBR Green PCR master mix (Life Technologies, Carlsbad, CA, USA) and ABI 7500 sequence detection system (Life Technologies). The real-time PCR primers: MRP2 forward: 5′-GCCAACTTGTGGCTGTGATAGG-3′; MRP2 reverse primer: 5′-ATCCAGGACTGCTGTGGGACAT-3′. MDR1 forward: 5′-GCTGTCAAGGAAGCCAATGCCT-3′; MDR1 reverse primer: 5′-TGCAATGGCGATCCTCTGCTTC-3′. RPL37A forward: 5′-AATCAGCCAGCACGCCAAGTAC-3′; RPL37A reverse primer: 5′-GCCACTGTCTTCATGCAGGAAC-3′. All primers were purchased from OriGene (Rockville, MD, USA) and checked for specificity using BLAST (NCBI). Exon/intron junctions were spanned.

### 2.7. Tissue Dissociation and Organoid Culture

Patient-derived organoids were isolated and cultured using minor modifications of a protocol previously published by Lee et al. [14]. Tumour tissues from patients were washed in PBS containing penicillin/streptomycin. Tumour tissues were minced with scissors and incubated in 10 mL of the organoid culture medium (hepatocyte medium with 10 ng/mL EGF, 5% charcoal-stripped FBS, 10 mM Y-27632, 100 mg/mL Primocin, and 1× Glutamax) supplemented with 1 mL collagenase/hyaluronidase (STEMCELL Technologies) at 37 °C for 15 min. Dissociated tissues were spun down at 300 g for 5 min, resuspended in 10 mL of PBS, and spun down again. The tissues were resuspended in 5 mL of TrypLE Express (Invitrogen, Life Technologies, Carlsbad, CA, USA) and incubated at room temperature for 3 min. Dissociated tissues were spun down at 300 g for 5 min, resuspended in 10 mL of HBSS supplemented with 5% charcoal-stripped FBS, 10 mM Y-27632 and 100 mg/mL Primocin, and passed through a 100 μm cell strainer. Dissociated cells (1 × 10^6^ cells/well) were spun down at 300 g for 5 min, resuspended in 60% Matrigel/organoid culture medium, plated in a 250 μL drop in the middle of one well of a pre-coated 6-well plate with 60% Matrigel, and solidified at 37 °C for 30 min. After solid drops formed, 1.5 mL of the organoid culture media was added to the well.

### 2.8. Organoid Drug Response Assay

Patient-derived organoids were collected after passaging and passed through a 100 μm cell strainer to eliminate large organoids. Subsequently, organoids were resuspended in 2% Matrigel/organoid culture medium (150–200,000 organoids/mL) in 15-mL polypropylene tubes and spun down at 300 g for 5 min. The bottom of the tubes was covered with ultrasound transmission gel. The EvoTron™ shock wave applicator was gently placed directly on the bottom of the tube. The shock wave frequency was four pulses per second. Organoids were exposed to 200 shock wave pulses at 0.12 mJ/mm^2^ and dispensed into ultralow-attachment 96-well plates (Corning Inc., Corning, NY, USA) in triplicate. At 24 h after plating, organoids were exposed to 0.16, 0.8, or 4 μM cisplatin for six days of drug incubation and cell viability was assayed by CellTiter-Glo 3D (Promega, Madison, WI, USA) according to the manufacturer’s instructions.

### 2.9. Statistical Analysis

Statistical analysis for each experiment is described in the figure legends. All graphs and analyses were calculated with GraphPad Prism 8.0.2 software (San Diego, CA, USA) and analyzed using one-way ANOVA followed by Tukey’s multiple-comparison test or Student’s *t*-test. Results are expressed as mean ± standard error of mean (SEM).

## 3. Results

### 3.1. Shock Waves Enhanced Cisplatin Cytotoxicity in UTUC Cells

We first determined the effect of shock wave treatment on the viability of BFTC909 cells and subsequently treated BFTC909 cells with 0, 50, 100, 150, 200, 250, 300, 400, 500, 800, or 1600 shock wave pulses at 0.05, 0.12, or 0.25 mJ/mm^2^. As shown in Figure 1A, the trypan blue dye exclusion assay indicated that the viability of BFTC909 cells remained 86% following shock wave treatment with 200 pulses at 0.12 mJ/mm^2^ or 83% following shock wave treatment with 50 pulses at 0.15 mJ/mm^2^. To explore the effects of shock waves in drug permeability into cells, BFTC909 cells were treated with 200 shock wave pulses at 0.12 mJ/mm^2^ or 50 pulses at 0.25 mJ/mm^2^ in the presence of Calcein and cellular uptake of Calcein was detected by flow cytometry. We found that BFTC909 cells treated with Calcein plus shock waves increased fluorescence compared to cells incubated with Calcein alone (Figure 1B). Therefore, we investigated whether shock waves could enhance the antitumour effect of cisplatin. BFTC909 and UM-UC-14 cells were treated with 3 μM cisplatin and shock waves as described above and their viability was evaluated using the MTT assay. Compared with control cells, BFTC909 and UM-UC-14 cells treated with cisplatin showed significantly decreased growth, which was further reduced by shock wave pulses (Figure 1C,D). Altogether, the results indicate that shock wave treatment increases the antitumour effect of cisplatin in vitro.

### 3.2. Shock Waves Improved Antitumour Effects of Cisplatin in BFTC909 and UM-UC-14 Xenografts

To explore whether shock waves could enhance the antitumour effect of cisplatin in vivo, we subcutaneously injected BFTC909 cells into immunodeficient SCID mice treated with cisplatin with or without shock wave treatment. The animals were divided into the following groups: (a) control mice, (b) control mice administered with LESW, (c) mice administered 1 mg/kg cisplatin, and (d) mice administered 1 mg/kg cisplatin with LESW. The experimental scheme is illustrated in Figure 2A. As shown in Figure 2B, an anaesthetised mouse was placed prostrate, and its tumour was placed in the focal area of the shock wave apparatus. Ultrasound transmission gel served as contact medium between the shock wave apparatus and the skin surface over the tumour. We did not observe any significant damage to the skin over the tumour after shock wave treatment (Figure 2C). We examined the effects of shock wave treatment on growth in the BFTC909 xenograft model and found that shock waves enhanced the antitumour effect of cisplatin in comparison with the effect observed with cisplatin treatment alone (Figure 2D–F). We observed similar results in the UM-UC-14 xenograft (Figure 2G–I).

### 3.3. Combination of Cisplatin and Shock Waves Additively Suppressed Tumour Cell Proliferation and Enhanced DNA Damage In Vivo

Haematoxylin and eosin (HE) staining was performed to examine the microscopic morphology of the BFTC909 xenograft tumour in each group and the morphological changes within the cells (Figure 3(A,top)). The control group (PBS) showed abundant tumour cells. Tumour tissues in the LESW alone group showed more vacuolisation, and those in the cisplatin alone group presented more lysis. In addition, HE staining revealed that tumour tissue structures were more numerous and loosely spaced in the LESW + cisplatin group. Further, Ki-67 immunostaining revealed that the proliferation of BFTC909 cells was suppressed following shock wave exposure and cisplatin administration (Figure 3((A,middle),B)). Ki-67 immunostaining in the control group did not significantly differ from that in the LESW group. Additionally, we observed that the expression of γ-H2AX was high in the cisplatin and shock wave treatment group (Figure 3((A,bottom),B)). The Combination of cisplatin and shock waves enhanced the activation of caspase-3 and caspase-9 apoptosis signaling in BFTC909 and UM-UC-14 cells (Appendix A). As expected, we observed similar results in the UM-UC-14 xenograft (Figure 3C,D). These results suggest that shock wave treatment improved the antitumour effect of cisplatin in BFTC909 and UM-UC-14 xenografts.

### 3.4. Shock Wave Treatment Enhanced Cisplatin Delivery into Tumour through Downregulation of E-Cadherin and ZO-1

Based on the aforementioned findings, we sought to confirm whether shock waves could promote cellular uptake of the chemotherapy drug cisplatin. Cisplatin is a chemotherapy agent that contains platinum and interacts with DNA in the form of the Pt-d(GpG) di-adduct, which triggers apoptosis in the affected cell. In this study, immunochemistry analysis of cisplatin–DNA adduct formation in the BFTC909 xenograft revealed that immunostaining of the cisplatin–DNA adduct was stronger in the LESW + cisplatin group than in the cisplatin alone group (Figure 4(A,top)). Cisplatin–DNA adduct formation was not detected in the control and LESW groups, as revealed by immunostaining. The results indicate that shock wave treatment enhances the uptake of cisplatin in vivo. In Figure 3A,C, we observed that tumour tissue structures were more loosely spaced in the LESW alone and LESW + cisplatin group, which may be caused by changes in the tight junction. In our previous study, we observed that LESW induced downregulation of ZO-1, a cytoplasmic plaque protein of tight junctions [20]. Our data showed that ZO-1 immunostaining was weaker in the LESW group and was the weakest in the LESW + cisplatin group in this study (Figure 4((A,middle),C)). In addition, we observed that LESW reduced the expression of the adherens junction protein E-cadherin (Figure 4((A,bottom),C)). As expected, we observed similar results in the UM-UC-14 xenograft (Figure 4B,D). Furthermore, the results indicated that effective silencing of ZO-1 or E-cadherin expression in BFTC909 and UM-UC-14 cells increases the permeability (Appendix A). Collectively, these results support that shock wave treatment facilitates cisplatin delivery by suppressing the expression of ZO-1 and E-cadherin.

### 3.5. Combination of Cisplatin and Shock Waves Repressed MDR1 Expression In Vivo

It has been previously reported that the cisplatin resistance of SSK2/R2 tumours was overcome by a transient increase in cisplatin uptake upon exposure to shock waves [21], and that MRP2 and MDR1 are important multidrug resistance transporter proteins for chemoresistance through the cellular efflux of cisplatin [22,23]. Therefore, in this study, we investigated whether LESW could regulate MRP2 and MDR1 expression in vivo. We found that LESW reduced the mRNA expression of MRP2 and MDR1 in the BFTC909 xenograft treated with cisplatin (Figure 5A). Moreover, we observed that MDR1 immunostaining was weaker in the LESW + cisplatin group than in the cisplatin alone group (Figure 5C). As expected, we observed a similar phenomenon in the UM-UC-14 xenograft model (Figure 5B,D). Taken together, the results revealed that LESW applied after the administration of cisplatin suppressed the mRNA transcription and protein expression of MDR1 in both BFTC909 and UM-UC-14 xenografts.

### 3.6. Shock Waves Improved Antitumour Effect of Cisplatin in Patient-Derived Organoid Model of UTUC

Several in vitro and in vivo studies have demonstrated that LESW enhances the antitumour effect of cisplatin. Moreover, in this study, we observed that the PDO model of UTUC could reflect the biological characteristics of the tumour tissue and drug efficacy. Our cultural conditions are similar to those described by Lee et al. [14] for human bladder cancer organoids. This study generated independent UTUC organoids (KCGMH-1–KCGMH-6) corresponding to six patients (Appendix A). The cells of the organoids were similar to their corresponding parental tumours according to HE staining and immunostaining for the indicated markers (Figure 6). The data showed high concordance in their histopathological and molecular features. To explore the effects of shock waves in drug permeability into organoids, KCGMH-02 and KCGMH-06 organoids were treated with 200 shock wave pulses at 0.12 mJ/mm^2^ in the presence of Lucifer yellow (LY) and tissue permeability of LY was detected by confocal microscopy. Tissue uptake of LY was clearly visible in organoids treated with LESW, in contrast, untreated organoids did not show LY fluorescence (Figure 7A). Additionally, KCGMH-02 and KCGMH-06 organoids were treated with 200 shock wave pulses at 0.12 mJ/mm^2^ in the presence of Calcein and detected by flow cytometry. We observed that organoids treated with Calcein plus shock waves increased fluorescence compared to organoids incubated with Calcein alone (Figure 7B). Hence, we explored whether the LESW improves cisplatin cytotoxicity in the PDO model of UTUC. First, the organoids were treated with LESW following 200 pulses at 0.12 mJ/mm^2^ on Day 0. On the next day, cisplatin was added for 6 days, followed by an evaluation using a 3D cell viability assay (Appendix A). In the PDO model, we showed that the combination of cisplatin and LESW could improve cisplatin cytotoxicity, except for the KCGMH-04 organoid (Figure 7C). It is speculated that this shock wave condition may be unsuitable for this specimen, which means that each tumour tissue sample has diffing sensitivity to shock waves. Thus, we observed that more sensitivity of the organoids to shock waves results in higher antitumour effects of the combination therapy. In summary, LESW could improve the antitumour effect of cisplatin in the preclinical PDO model.

## 4. Discussion

Extracorporeal shock wave therapy is currently used to treat many diseases. It involves the use of a focused acoustic wave that carries energy to the treatment site. High-energy shock waves have been used for the disintegration of urinary tract stones since 1980. Subsequently, several studies have reported the application of LESW to human tissue for the regeneration or repair of bones [24], muscles [25], and other soft tissues [26,27]. The advantages of shock waves are their noninvasiveness, ease of handling by clinicians, and the short-term treatment course.

Several studies have described the possible role of LESW in cancer treatment [28,29,30]. LESW induces gene transfection through secreted microparticles [31]. Some studies have focused on improving the focal permeability of macromolecular cytotoxic drugs into specific cancer tissue [32]. For example, some studies have reported that LESW combined with chemotherapy can improve the treatment efficacy of chemotherapy by increasing the intracellular drug concentration [28,33]. In this study, we observed a similar result in vitro and in vivo and further identified decreased expression of the tight junction protein ZO-1 and the adherent junction protein E-cadherin when chemotherapy was used in combination with LESW. This combination treatment facilitated the infiltration of the cytotoxic agent more deeply into the cancer tissue, improving the overall anticancer efficacy.

Weiss et al. found that LESW can improve the efficacy of cisplatin in the cisplatin-resistant cell line SSK2/R2 [21]. Cisplatin is currently the main treatment drug for advanced urothelial carcinoma. In our study, we further explored the effect of LESW on membranous cisplatin transport proteins such as MDR1 and MRP2. We observed that MDR1 expression decreased after the use of LESW. This mechanism explains why LESW causes the increase in the intracellular cisplatin level and overcomes any potential cisplatin resistance to enhance tumour sensitivity to chemotherapy. Therefore, as per our preliminary result, LESW can improve the anticancer effect through two mechanisms. First, it improves the infiltration of chemotherapy drugs into the deep cancer tissue by interfering with ZO-1 and E-cadherin expression, resulting in changes to their permeability; second, it increases the intracellular drug level by reducing the expression of cisplatin efflux membranous protein (MDR-1).

The organoid model used in this study is widely applied for cancer research, as it represents the parental tumour with several molecular characteristics [14,15,16,17,34]. Vlachogiannis et al. [11] proved that the organoid cancer model can be used to predict treatment response or in clinical situations (even after a complicated treatment strategy). In this study, the organoid model exhibited histology and protein expression similar to those of the parental tumour. The synergic effect of LESW on the antitumour effect of cisplatin is compatible with in vitro and animal models. To the best of our knowledge, this is the first study to use the organoid model to test the effect of LESW on cancer tissue. In addition to the animal model, the preclinical organoid model further supported the possible role of LESW combined with chemotherapy in human cancer treatment.

In this study, LESW alone did not promote tumour growth in UTUC cell lines, animal models, and PDO models. The combination of LESW and cisplatin exerted higher antitumour effects. However, different cell lines and PDO models showed varying antitumour effects, which could be attributed to different cell sizes, different components of cell membranous protein, individualised cytoskeleton structure, and uncontrolled structural constituents of the extracellular matrix. Therefore, the optimal frequency and intensity of LESW might differ in different cell lines or PDO models. LESW is widely used in humans for translational purposes, without any obvious side effects. The efficacy of cisplatin must be improved for the treatment of UTUC because most patients with UTUC are older and may have renal insufficiency. The inadequate dose of cisplatin consequently leading to a poor outcome is a clinical challenge. Through this study, we provide an encouraging result about the synergistic role of LESW in the efficacy of cisplatin in an in vitro study and a preclinical PDO model. The application of LESW in humans and the clinical benefits of the treatment warrant further investigation.

This study provided the preliminary encouraging result of the synergistic antitumour effect of LESW and cisplatin. However, there were still some limitations that should be disclosed. The systemic tumour seeding during LESW manipulation is a major concern during anticancer treatment. For the same reason, the perioperative chemotherapy strategy is commonly used for the prevention of systemic circulating tumour cell seeding during surgical manipulation. The circulating tumour cell resistance to perioperative chemotherapy is associated with disease recurrence and oncological outcome [35]. In this study, we identified the decreased tumour activity and cisplatin resistance in the group of LESW-Cisplatin combination treatment. We advised that LESW alone should not be considered for real-world cancer patients but a combination strategy can eliminate the concern of systemic cancer cell spreading but also improved the local tumour response. For clinical translational purposes, the application human trial of LESW on UTUC treatment should be performed in the strict protocol. First, LESW should be combined with adequate systemic chemotherapy under closed circulating tumour cell activity monitoring. Second, LESW is not suitable for disseminated metastasis but can be considered for locally advanced UTUC to improved neoadjuvant treatment response rate or for those patients with symptomatic solitary metastasis. Both the above two situations are clinical unmet needs for cisplatin-resistant urothelial carcinoma. The role of reversed cisplatin resistance mechanism of LESW in the study seems to offer an alternative treatment choice. Third, the precise tumour targeting technique can be overcome by a fiducial marker commonly used in uro-oncological treatment [36]. We look forward to a careful study design to explore the potential benefit of LESW-Cisplatin combination treatment based on this preclinical finding in the near future.

## 5. Conclusions

Our study demonstrated the feasibility of the combination of LESW and cisplatin for treating UTUC through in vitro, in vivo, and PDO models. LESW can improve the infiltration of cisplatin into deep cancer tissue by interfering with ZO-1 and E-cadherin, causing subsequent changes to their permeability, and increasing the intracellular drug level by reducing the expression of the cisplatin efflux membranous protein MDR-1 (Figure 8). This study is the first to prove that noninvasive shock waves improve the cytotoxicity of cisplatin in a preclinical PDO model of UTUC. Further studies that focus on optimisation of the treatment efficacy of this combination are warranted.

## Figures and Tables

**Figure 1 cancers-13-04558-f001:**
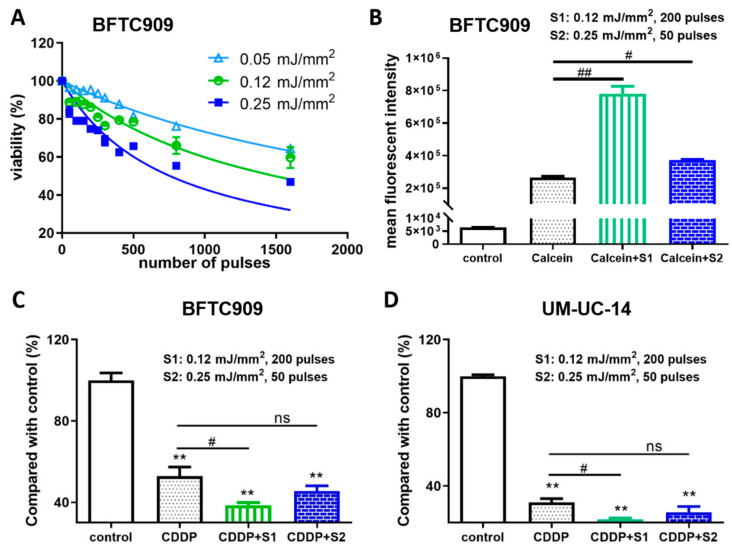
Shock wave treatment enhanced the antitumour effect of cisplatin (CDDP) in vitro. (**A**) Determination of optimal conditions of shock waves for human UTUC BFTC909 cell line. BFTC909 cells were treated with the indicated number of shock wave pulses at 0.05, 0.12, and 0.25 mJ/mm^2^. (**B**) BFTC909 cells were treated with 0.2 mM Calcein combined with 200 shock wave pulses at 0.12 mJ/mm^2^ (S1) or 50 pulses at 0.25 mJ/mm^2^ (S2) and Calcein permeability was assessed by flow cytometry. Data are denoted as mean ± SEM, *n* = 3, the *p* values were calculated with one-way ANOVA followed by Tukey’s multiple-comparison test, # *p* < 0.05 versus Calcein group, ## *p* < 0.01 versus Calcein group. BFTC909 (**C**) and UM-UC-14 (**D**) cells were treated with 3 μM cisplatin combined with shock wave treatment as described. After 72-h treatment, cell proliferation was assessed using the MTT assay. Data are denoted as mean ± SEM, *n* = 6, the *p* values were calculated with one-way ANOVA followed by Tukey’s multiple-comparison test, ** *p* < 0.01 versus control group, # *p* < 0.05 versus CDDP group, ns indicates no significance.

**Figure 2 cancers-13-04558-f002:**
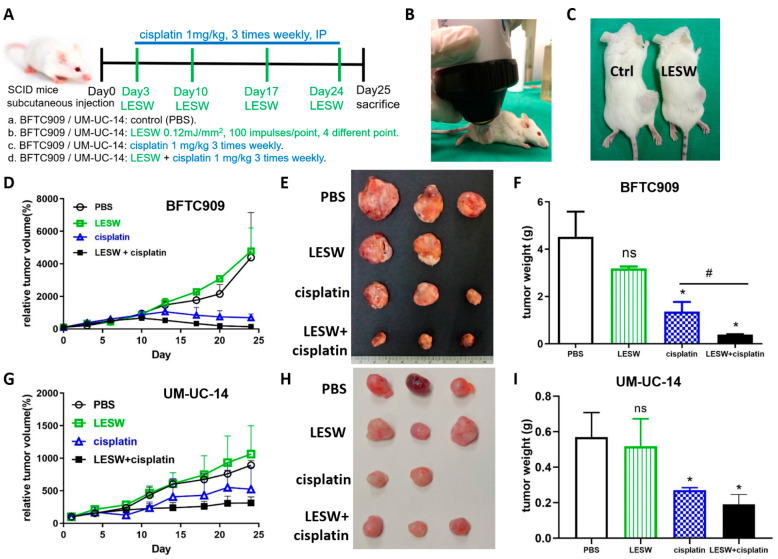
Shock wave treatment augmented cisplatin effects in vivo. (**A**) Flowchart showing the timeline and treatment methods of the experiment. (**B**) Experimental apparatus for in vivo shock wave exposure. The shock waves were produced by Duolith SD-1 (Storz Medical, Tagerwilen, Switzerland). The xenograft tumour was directly exposed to shock wave treatment through the ultrasound transmission gel. (**C**) Shock waves did not cause significant damage to the skin over the tumour. Relative tumour volumes of BFTC909 (**D**) and UM-UC-14 (**G**) xenografts in SCID mice after treatment for 25 days; the initial tumour volumes reached 100–150 mm^3^. Tumour diameters were measured twice a week using vernier callipers. BFTC909 (**E**) and UM-UC-14 (**H**) representative tumours in each group. The mass of BFTC909 (**F**) and UM-UC-14 (**I**) tumours was weighed after the mice were sacrificed. Error bars represent mean ± S.E.M. The statistical test performed was one-way-ANOVA, * *p* < 0.05 versus control group (PBS), # *p* < 0.05 versus CDDP group, ns indicates no significance.

**Figure 3 cancers-13-04558-f003:**
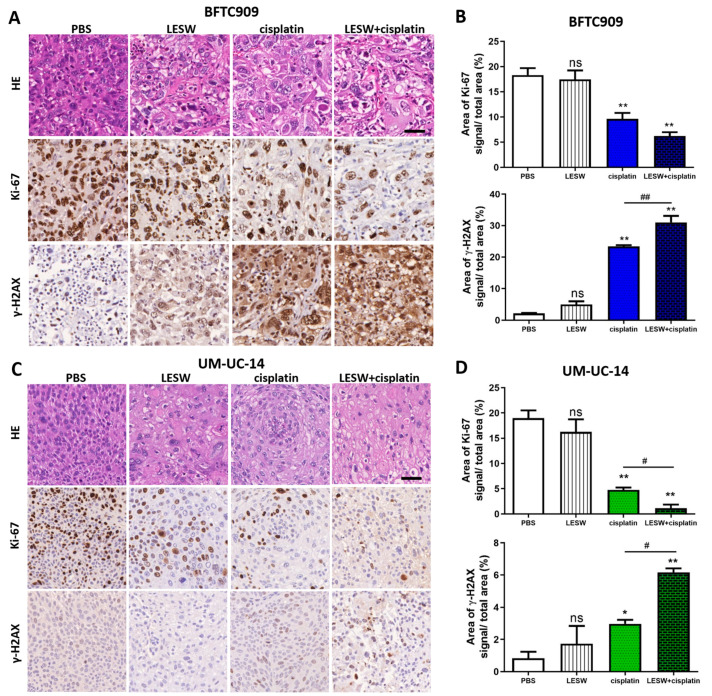
Histopathologic characteristics and immunohistochemical analyses of Ki-67 and γ-H2AX in UTUC cell line–derived tumour xenografts. HE staining and immunostaining of Ki-67 and γ-H2AX in BFTC909 (**A**) and UM-UC-14 (**C**) xenograft tumours in SCID mice. Immunodetectable proteins were stained brown, and the nuclei were counterstained blue. Scale bars indicate 40 μm. Statistical analysis of the percentage of Ki-67 and γ-H2AX signal area of the four groups in BFTC909 (**B**) and UM-UC-14 (**D**) xenografts. Error bars represent mean ± S.E.M., * *p* < 0.05, ** *p* < 0.01 versus control group; ns indicates no significance, # *p* < 0.05 versus CDDP group, ## *p* < 0.01 versus CDDP group.

**Figure 4 cancers-13-04558-f004:**
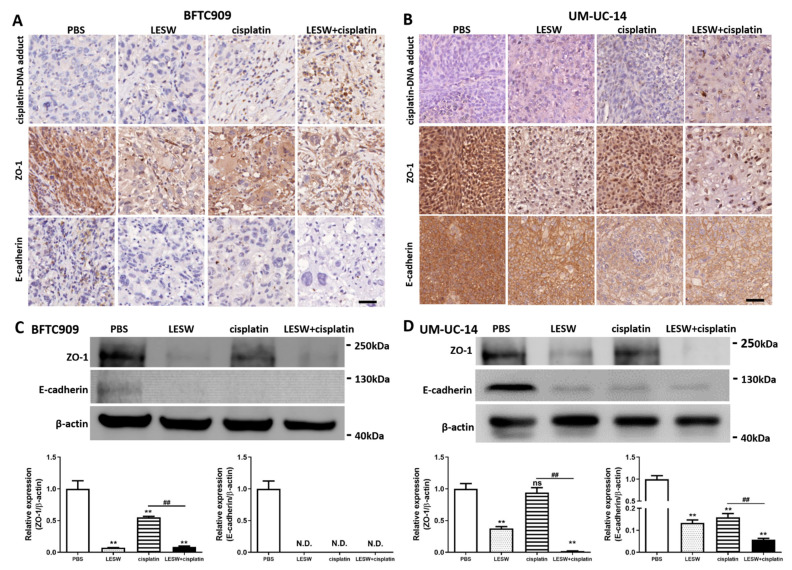
LESW suppressed E-cadherin and ZO-1 expression to increase cisplatin delivery in vivo. Immunohistochemical analysis of BFTC909 (**A**) and UM-UC-14 (**B**) xenograft tumour tissue sections immunolabeled for cisplatin–DNA adduct (top), ZO-1 (middle) and E-cadherin (bottom). Scale bars indicate 40 μm. The lysates from BFTC909 (**C**) and UM-UC-14 (**D**) xenograft tumour were analysed by Western blotting with the indicated antibodies. Error bars represent mean ± S.E.M., N.D.: not detected, the *p* values were calculated with one-way ANOVA followed by Tukey’s multiple-comparison test, ** *p* < 0.01 versus control group, ns indicates no significance between the control group and CDDP group, ## *p* < 0.01 versus CDDP group. Full Western blot images can be found in Appendix A.

**Figure 5 cancers-13-04558-f005:**
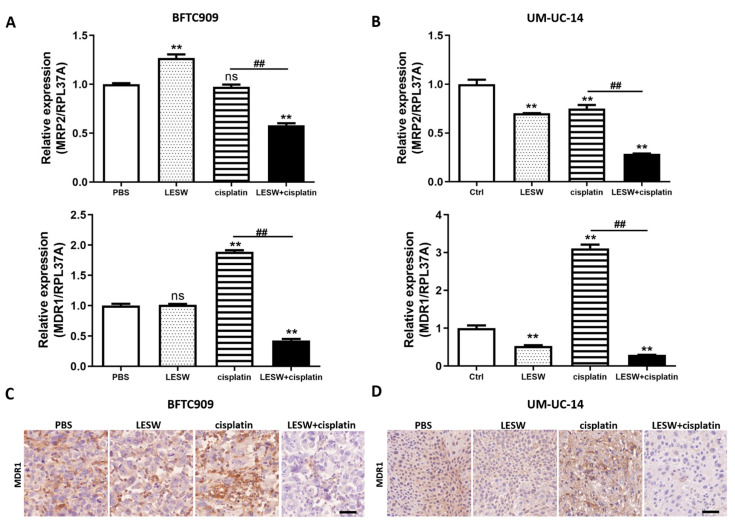
Combined treatment with LESW and cisplatin downregulated MDR1 expression in UTUC cell line xenograft models. (**A**,**B**) Quantitative polymerase chain reaction analysis of MRP2 and MDR1 expression in BFTC909 (**A**) and UM-UC-14 (**B**) xenografts. Error bars represent mean ± S.E.M. The statistical test performed was one-way-ANOVA, ** *p* < 0.01 versus control group; ns indicates no significance, ## *p* < 0.01 versus CDDP group. (**C**,**D**) Immunohistochemical imaging of BFTC909 (**C**) and UM-UC-14 (**D**) xenograft tumour tissue sections immunolabeled for MDR1. Scale bars indicate 40 μm.

**Figure 6 cancers-13-04558-f006:**
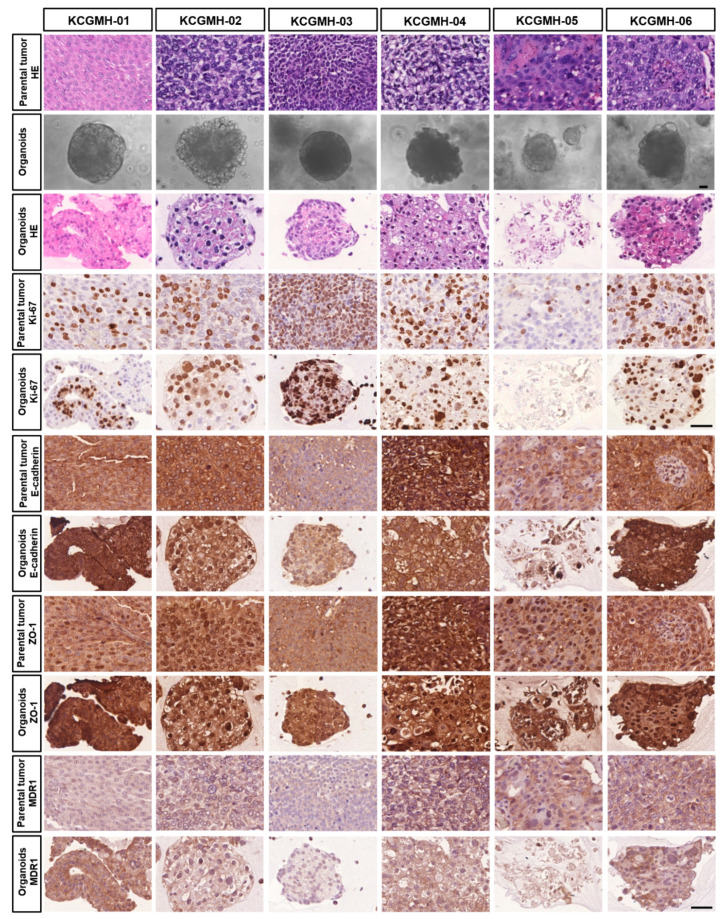
Establishment of Patient-Derived Tumor Organoids of UTUC. Bright-field images of organoids together with H&E staining and immunostaining for the indicated markers of parental tumors and patient-derived organoids. Scale bars indicate 50 µm.

**Figure 7 cancers-13-04558-f007:**
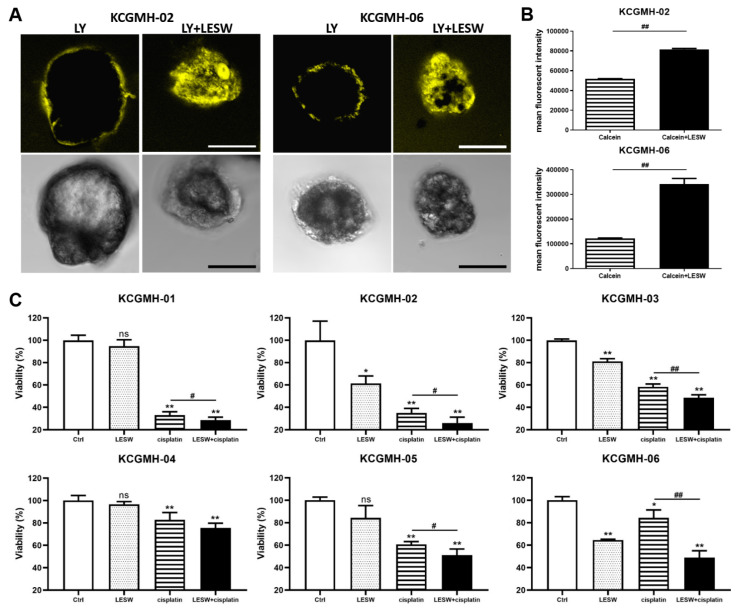
Shock waves enhanced cisplatin cytotoxicity in a patient-derived organoid (PDO) model of UTUC. (**A**) Confocal fluorescent images for KCGMH-02 and KCGMH-06 organoids in the presence of Lucifer yellow (LY) treated with or without 200 shock wave pulses at 0.12 mJ/mm^2^. (yellow: Lucifer yellow, size bar = 20 μm). (**B**) KCGMH-02 and KCGMH-06 organoids were treated with 0.2 mM Calcein combined with 200 shock wave pulses at 0.12 mJ/mm^2^ and assessed by flow cytometry. Data are denoted as mean ± SEM, *n* = 3, the *p* values were calculated with Student’s t-test, ## *p* < 0.01 versus Calcein group. (**C**) Combined treatment with cisplatin and LESW augmented the effect of cisplatin on cell viability in the PDO model. Six independent PDOs were treated with 200 shock wave pulses at 0.12 mJ/mm^2^, cisplatin, or 200 shock wave pulses at 0.12 mJ/mm^2^ plus cisplatin. KCGMH-01 was treated with 4 μM cisplatin, and the other PDOs were treated with 0.16 μM cisplatin. Cell viability was measured using the CellTiter-Glo assay after six days of cisplatin treatment. Values are mean ± S.E.M. of three biological replicates. Error bars represent the mean ± S.E.M., the *p* values were calculated with one-way ANOVA followed by Tukey’s multiple-comparison test, * *p* < 0.05, ** *p* < 0.01 versus control group, ns indicates no significance, # *p* < 0.05 versus CDDP group, ## *p* < 0.01 versus CDDP group.

**Figure 8 cancers-13-04558-f008:**
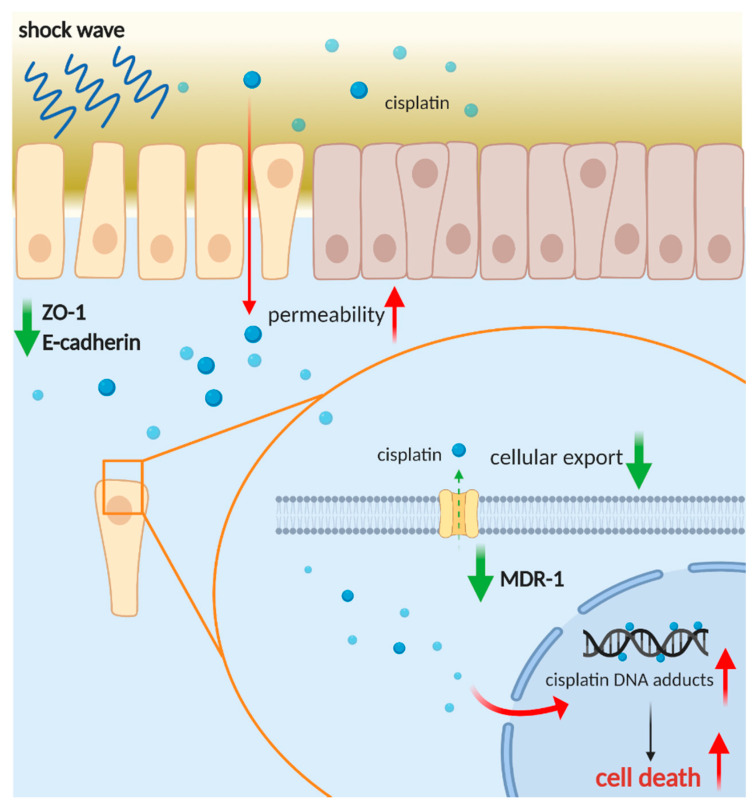
Schematic representation of the role of shock wave treatment in improving infiltration and anticancer effect of cisplatin. Created with https://biorender.com (accessed on 30 March 2021).

## Data Availability

Luo and Chuang had full access to all the data in the study and take responsibility for the integrity of the data and the accuracy of the data analysis. The data presented in this study are available on request from the corresponding author. The data are not publicly available due to privacy restrictions.

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
