# Peer review of "Extracorporeal Shock Wave Enhances the Cisplatin Efficacy by Improving Tissue Infiltration and Cellular Uptake in an Upper Urinary Tract Cancer Animal and Human-Derived Organoid Model"

_cancers, 2021, doi:10.3390/cancers13184558_

Round 1

Reviewer 1 Report

No further request for this article. 

Reviewer 2 Report

In this revised version, the authors clarified molecular mechanisms by which ESW enhances drug permeability to enhance antitumor effects of CDDP, which is convincing to the readers. The reviewer would like to congratulate the authors!

This manuscript is a resubmission of an earlier submission. The following is a list of the peer review reports and author responses from that submission.

Round 1

Reviewer 1 Report

The author presented the role of LESWs in the treatment of prostate cancer. The result was significant with the mechanism of the Zo-1/E cadherin-related pathway.  The work is original and interesting. 

Major 

Since most of the chemotherapy will trigger apotosis. I would like to see the difference in the apoptotic pathway through the FACS or Western blotting related to the Caspase-9 or Caspase -3.  

Reviewer 2 Report

The authors demonstrated an additive effect of ESW on anti-tumor effects of CDDP in human urothelial cancer in virro and in vivo. However, the present study is short in terms of molecular mechanisms. The authors may want to address the following point.  

The authors described that ESW facilitated CDDP delivery by suppressing expression of ZO-1 and E-cadherin. The authors just observed downregulation of ZO-1 and E-cad in conditions at which ESW exhibited anti-tumor effects additional to those induced by CDDP. Although the authors mentioned that ESW facilitates the infiltration of CDDP into the deep cancer tissue by interfering with ZO-1 and E-cad (lines 384-6), the downregulation of ZO-1 and E-cad may be independent of the additional anti-tumor effects of ESW. The authors may want to show that molecular abrogation of ZO-1 and E-cad using siRNA etc. exhibits the similar anti-tumor effects induced by ESW in combination with CDDP as well as an increase in the CDDP permeability into UC cells.

Reviewer 3 Report

- As a clinical principle, any manipulation on urothelial cancer must be avoided as it has a high risk of seeding. This would be the case with ESWL. In my opinion, this precluldes any further testing. Was seeding investigated? 

- In general, patients with metastatic upper tract urothelial carcinoma have more than one site of metastatic disease. How would this work in clinical practice? Treat all sites? Also, how does one target the metastasis? ESWL is generally done under Xray or ultrasound guidance, but the majority of metastasis may not be visible with these imaging techniques. 

- Cisplatin alone is not given in metastatic urothelial carcinoma, but a combination with Gemcitabine or Vinblastin/Adriamycin. It may be reasonable to test these drugs. 

- There have been a number of IO drugs approved for the treatment of metastatic upper tract urothelial carcinoma. Consider testing them.